 

# Dnmt3a is an epigenetic mediator of adipose insulin resistance

Dongjoo You[1], Emma Nilsson[2], Danielle E Tenen[3], Anna Lyubetskaya[4], James C Lo[5], Rencong Jiang[1], Jasmine Deng[1], Brian A Dawes[3], Allan Vaag[6,7], Charlotte Ling[2], Evan D Rosen[3,4]*, Sona Kang[1]*

[1]Nutritional Sciences and Toxicology Department, University of California, Berkeley, Berkeley, United States; [2]Epigenetics and Diabetes Unit, Department of Clinical Sciences, Lund University Diabetes Centre, Scania University Hospital, Malmö, Sweden; [3]Division of Endocrinology, Diabetes and Metabolism, Beth Israel Deaconess Medical Center, Boston, United States; [4]Broad Institute of Harvard and MIT, Cambridge, United States; [5]Weill Cornell Medical College, New York, United States; [6]Diabetes and Metabolism, Department of Endocrinology, Rigshospitalet, University of Copenhagen, Copenhagen, Denmark; [7]Early Clinical Development, AstraZeneca, Innovative Medicines, Göteborg, Sweden

*For correspondence:
erosen@bidmc.harvard.edu (EDR);
kangs@berkeley.edu (SK)

Competing interests: The authors declare that no competing interests exist.

**Abstract** Insulin resistance results from an intricate interaction between genetic make-up and environment, and thus may be orchestrated by epigenetic mechanisms like DNA methylation. Here, we demonstrate that DNA methyltransferase 3a (Dnmt3a) is both necessary and sufficient to mediate insulin resistance in cultured mouse and human adipocytes. Furthermore, adipose-specific Dnmt3a knock-out mice are protected from diet-induced insulin resistance and glucose intolerance without accompanying changes in adiposity. Unbiased gene profiling studies revealed *Fgf21* as a key negatively regulated Dnmt3a target gene in adipocytes with concordant changes in DNA methylation at the *Fgf21* promoter region. Consistent with this, Fgf21 can rescue Dnmt3a-mediated insulin resistance, and DNA methylation at the *FGF21* locus was elevated in human subjects with diabetes and correlated negatively with expression of *FGF21* in human adipose tissue. Taken together, our data demonstrate that adipose Dnmt3a is a novel epigenetic mediator of insulin resistance in vitro and in vivo.
DOI: https://doi.org/10.7554/eLife.30766.001

## Introduction

Insulin resistance (IR) is generally defined as a reduced ability of insulin to provoke metabolic responses, such as glucose uptake. Several mechanistic drivers of IR have been proposed, including defects in the insulin signal transduction pathway, dysfunctional mitochondrial respiration, and ER stress (*Boucher et al., 2014*; *Gregor and Hotamisligil, 2011*; *Houstis et al., 2006*; *Ozcan et al., 2004*; *Samuel and Shulman, 2012*). The data supporting the involvement of these non-mutually exclusive pathways are convincing, but additional mechanisms are also likely to participate in the development of IR. Nuclear pathways in particular (i.e. transcriptional and epigenomic mechanisms) are likely to play an important role, given that IR often develops over a time course that is inconsistent with rapid signal transduction events (*Kang et al., 2015*; *Tan et al., 2015*). Furthermore, there is a large body of data from humans and rodents indicating that the propensity for IR can be passed transgenerationally; such 'fetal programming' studies strongly imply an epigenetic basis for IR (*Carone et al., 2010*; *Heijmans et al., 2008*; *Li et al., 2010*; *Tobi et al., 2014*). Finally, thiazolidinedione drugs, which activate the transcription factor PPARγ, provide one of the only pharmacological therapies for IR (*Soccio et al., 2014*; *Step et al., 2014*; *Sugii et al., 2009*). Taken together, these

data strongly suggest that epigenomic and transcriptional mechanisms must be in play in the etiopathogenesis of IR (*Kang et al., 2016*).

DNA methylation is a reversible epigenetic mark involving the covalent transfer of a methyl group to the C-5 position of a cytosine residue, usually in the context of a CpG doublet, by DNA methyltransferases (Dnmts) (*Robertson, 2005*). Long thought to be a static epigenetic mark, emerging evidence suggests that methylated DNA undergoes dynamic and reversible remodeling in somatic cells during developmental, physiological, and pathogenic pathophysiological processes (*Barres et al., 2013*; *2009*; *Benner et al., 2015*; *Kirchner et al., 2016*; *Multhaup et al., 2015*; *Siersbæk et al., 2017*). As an example, a bout of intense exercise can induce DNA hypomethylation at the *PPARGC1a* locus in human skeletal muscle within hours, followed by increased gene expression; these effects were reversed after resting (*Barrès et al., 2012*). Mounting evidence points to a role for DNA methylation in the pathogenesis of metabolic disorders. For example, the obesity-prone *agouti* mouse has reduced DNA methylation at the regulatory region for the *Agouti* gene, leading to enhanced expression and subsequent inhibition of the satiety-inducing MC4 receptor (*Xie et al., 2013*). Multiple studies have shown that changes in DNA methylation at key metabolic genes, such as *Cox7a1* (*Rönn et al., 2008*), *IGF-2* (*Heijmans et al., 2008*), and *Pomc* (*Ehrlich et al., 2010*), associate with various metabolic insults including aging, obesity, anorexia, and prenatal exposure to famine. Moreover, recent genome-wide profiling studies have identified distinct global DNA methylation patterns that associate with obesity and diabetes in humans (*Dayeh et al., 2014*; *Feinberg et al., 2010*; *Kirchner et al., 2016*; *Multhaup et al., 2015*; *Nilsson et al., 2015*; *Volkmar et al., 2012*; *Volkov et al., 2017*). Altered DNA methylation is linked to the transgenerational passage of metabolic disorders (*Ng et al., 2010*; *Wei et al., 2014*). Despite numerous studies showing a link between DNA methylation and metabolic dysregulation, the cause-and-effect relationship between the two remains largely unknown.

In mammals, five Dnmt family members have been identified: *Dnmt1*, *Dnmt2* (also known as *Trdmt1*), *Dnmt3a*, *Dnmt3b* and *Dnmt3l*, yet only three (*Dnmt1*, *3a*, and *3b*) possess DNA methyltransferase activity (*Robertson, 2005*). Dnmt1 (the maintenance DNA methyltransferase) has a preference to methylate hemimethylated DNA whereas Dnmt3a and 3b prefer unmethylated DNA as substrate, and thus act as de novo DNA methyltransferases (*Robertson, 2005*). Dnmt3l is homologous to the other Dnmt3s but lacks catalytic activity (*Robertson, 2005*) and *Dnmt2* has sequence homology to all Dnmts but methylates cytoplasmic tRNA, and not DNA (*Robertson, 2005*). Administration of a small molecule Dnmt inhibitor has been shown to improve insulin sensitivity in the setting of obesity, in part by demethylating the *Adipoq* promoter, and a role for Dnmt1 in this process was suggested by genetic knockdown studies in cultured adipocytes (*Kim et al., 2015*).

Here, we identify Dnmt3a as a key epigenetic determinant of obesity-associated IR in adipose tissue. Dnmt3a plays a causal role in the development of cell autonomous IR in mouse and human adipocytes in a manner that is dependent on its catalytic activity. Consistent with results from adipocytes in vitro, we find that adipose-specific Dnmt3a deficiency in vivo confers protection from diet-induced IR and glucose tolerance without accompanying changes in body weight or adiposity. In addition, unbiased gene expression profiling suggests that Dnmt3a acts as primarily as a gene repressor, and reveals a set of adipocyte-specific target genes, including *Fgf21*. Further, we demonstrate that Dnmt3a mediates IR, at least in part, by methylating specific *cis*-regulatory elements in the *Fgf21* gene and thus suppressing its expression. We also provide evidence that similar mechanisms might be at play in human adipose tissue. These studies identify Dnmt3a as a novel epigenetic mediator of IR in adipocytes.

## Results

### Adipose expression of Dnmts inversely correlates with insulin sensitivity

To identify a potential role for Dnmts in the pathogenesis of insulin resistance, we compared their expression levels in insulin-sensitive metabolic tissues (i.e. liver, muscle, and adipose depots) in various metabolic settings. In a diet-induced obesity (DIO) mouse model, *Dnmt1* and *Dnmt3a* mRNA expression was elevated in adipose tissue, especially in epididymal and mesenteric WAT (*Figure 1A*, *Figure 1—figure supplement 1A*). Dnmt3a expression was significantly induced by HFD in skeletal

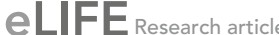

**Figure 1.** Increased adipose expression of Dnmts in obesity. (A) *Dnmt* mRNA expression in adipose from chow- and high fat-fed (HFD) C57Bl/6 mice (A); n = 6 mice, p<0.05, Student's *t*-test, mean ±s.e.m.) and (B) in adipose tissue from *ob/+*and *ob/ob* mice in the presence and absence of rosiglitazone treatment (right; n = 7 for *ob/+*, n = 8 for *ob/+*plus Rosi, n = 7 for *ob/ob* with and without Rosi, p<0.05, Student's *t*-test, mean ±s.e.m.). (C) Western blot showing Dnmt3a protein levels in adipose tissue from chow vs. HFD mice. (D) Quantification of western blot in (C).

DOI: https://doi.org/10.7554/eLife.30766.002

The following source data and figure supplements are available for figure 1:

**Source data 1.** Figure 1—source data.
DOI: https://doi.org/10.7554/eLife.30766.005
**Figure supplement 1.** Expression of Dnmts in various metabolic tissues.
DOI: https://doi.org/10.7554/eLife.30766.003
**Figure supplement 1—source data 1.** Figure 1—figure supplement 1—source data.
DOI: https://doi.org/10.7554/eLife.30766.004

muscle, but to a lesser degree than in adipose tissue. No effect of HFD was observed in liver (*Figure 1—figure supplement 1B–C*). In genetically obese *ob/ob* mice, mRNA expression of all three functional *Dnmts* was elevated in adipose tissue (*Figure 1B*). Importantly, expression of these genes was significantly reduced after treatment with the insulin sensitizer rosiglitazone (Rosi) (*Figure 1B*). Increased adipose expression of Dnmt3a was confirmed by immunoblotting in DIO mice (*Figure 1C, D*). Relative expression of Dnmt3a was higher in eWAT compared to skeletal muscle and liver compared in normal chow-fed mice (*Figure 1—figure supplement 1D–E*). *Dnmt2* and *Dnmt3l* mRNA displayed minimal expression in adipose tissue (not shown). Together, this led us to hypothesize that Dnmts may have a direct functional role in the pathogenesis of adipose IR.

## Knock-down of Dnmt3a confers protection from drug-mediated IR

Given the inverse correlation of adipose *Dnmt* expression with insulin sensitivity, we sought to test the cell autonomous role of Dnmts in the development of IR. To that end, individual Dnmts were knocked-down in mature 3T3-L1 adipocytes using two distinct short hairpin RNAs (shRNAs) (*Figure 2—figure supplement 1A–F*). To promote IR, cells were treated with dexamethasone (Dex) and TNF-$\alpha$ (TNF) (*Houstis et al., 2006*; *Kang et al., 2015*) for an additional four days. Knock-down of individual Dnmts did not have a major impact on the basal level of insulin sensitivity when assessed by insulin-stimulated glucose uptake (*Figure 2—figure supplement 2A–E*). However, knock-down of *Dnmt3a*, but not *Dnmt1* or *Dnmt3b*, protected adipocytes from Dex- and TNF-mediated IR as assessed by insulin-stimulated glucose uptake and signal transduction (*Figure 2A,B*, *Figure 2—figure supplement 2A–E*). Given the elevated expression of adipose Dnmts in obesity, we were prompted to test whether overexpression of Dnmts per se is sufficient to drive IR in adipocytes. Individual Dnmts were expressed in mature 3T3-L1 adipocytes by lentiviral delivery. Interestingly, overexpression of Dnmt3a and 3b, but not Dnmt1, potently inhibited insulin-stimulated glucose uptake (*Figure 2C*). Additionally, we tested whether the causal role of Dnmt3a in the development of insulin resistance is conserved in human adipocytes. Overexpression of Dnmt3a powerfully inhibited insulin-stimulated glucose uptake in in vitro differentiated adipocytes from human adipose derived stem cells (*Figure 2D*). Together, our gain- and loss-of-function studies in vitro show that Dnmt3a is both necessary and sufficient to mediate IR in adipocytes in a cell autonomous manner.

## DNA methyltransferase activity is critical for Dnmt3a to mediate IR

Studies have shown that Dnmt3a can act as a gene repressor using both catalytic-dependent and independent mechanisms (*Bachman et al., 2001*; *Fuks et al., 2001*). Thus, we sought to determine whether DNA methyltransferase activity is required to mediate IR. We initially pursued a pharmacological approach, using two synthetic Dnmt inhibitors, 5-azacytidine (5-Aza) (*Stresemann and Lyko, 2008*), a nucleoside analogue serving as a pseudo-substrate for Dnmts, and RG-108 (*Brueckner et al., 2005*), a non-nucleoside-based drug. Remarkably, both Dnmt inhibitors blocked Dex- and TNF-mediated IR (*Figure 3A,B*, *Figure 3—figure supplement 1A,B*). In addition to these synthetic inhibitors, the naturally occurring Dnmt inhibitors (*Rajavelu et al., 2011*) epigallocatechin gallate (ECCG) and theaflavin-3,3′-digallate (TF-3) also showed insulin-sensitizing activity in the setting of Dex- and TNF-mediated IR (*Figure 3—figure supplement 1C–F*). These data suggest that DNA methyltransferase activity is necessary for Dnmt3 to cause IR. To further probe this hypothesis, we overexpressed a catalytically inactive allele of Dnmt3a (C706S) (*Hsieh, 1999*). Unlike the wild-type allele, Dnmt3a C706S was unable to suppress insulin-stimulated glucose uptake (*Figure 3C*). Together, the pharmacological and genetic evidence indicates that DNA methyltransferase activity is essential for Dnmt3a to mediate IR.

## Adipose-specific Dnmt3a deficiency confers protection from diet-induced insulin resistance and glucose intolerance

We next sought to determine whether loss of Dnmt3a in adipocytes could confer protection from diet-induced adipose tissue and systemic IR in vivo. To address this, we generated adipose-specific Dnmt3a knock-out mice (*Dnmt3a^AdiKO*) by crossing Dnmt3a floxed mice (*Dnmt3a^f/f*) (*Nguyen et al., 2007*) to adiponectin-Cre mice (*Eguchi et al., 2011*) (*Figure 4—figure supplement 1A,B*). On chow diet, male WT and *Dnmt3a^AdiKO* animals did not show any obvious change in their body weight, insulin sensitivity, glucose tolerance, or insulin levels (*Figure 4—figure supplement 2A–D*). On HFD (45% fat), *Dnmt3a^AdiKO* mice still showed no significant differences in body weight or composition relative to littermate controls (*Figure 4A,B*). Moreover, the effect of long-term fasting (16 hr) on body weight or body composition did not differ between genotypes (*Figure 4C*), suggesting that gross energy homeostasis is not altered in the knock-out animals. Despite no obvious change in energy balance, high fat fed male *Dnmt3a^AdiKO* mice exhibited improved glucose tolerance (*Figure 4D,E*) and insulin sensitivity (*Figure 4F,G*) with reduced serum insulin in the fed state (*Figure 4H*). Consistent with their metabolic improvement, male *Dnmt3a^AdiKO* mice display increased insulin signal transduction in adipose tissue, as demonstrated by increased insulin-stimulated phosphorylation of IRS and AKT (*Figure 4I,J*). Similarly, female *Dnmt3a^AdiKO* mice display improved insulin sensitivity and glucose tolerance on HFD without a change in body weight (*Figure 4—figure*



**Figure 2.** Dnmt3a is sufficient and necessary to mediate cell autonomous IR in vitro. (A) Two independent hairpins against Dnmt3a (versus scrambled control shRNA; shScr) were delivered to mature 3T3-L1 adipocytes (Day 8) via lentiviral transduction. Cells were then treated with Dex and TNF for 4 days and assessed for insulin-stimulated glucose uptake ($^3$H-2-DG assay, n = 6, p<0.05). Shown is the % of insulin-stimulated glucose uptake rescued by *Dnmt3a* knockdown (n = 6, p<0.05). (B) Cells in (A) with #2 shDnmt3a hairpin and control cells were withdrawn from serum for 6 hr, stimulated with 20 nM insulin for 5 min and subjected to immunoblotting with antibodies against total and phospho-AKT and –IRS-1 with and without insulin. (C) Basal and insulin-stimulated glucose uptake in mature 3T3-L1 and (D) in in vitro-differentiated primary human adipocytes transduced with lentivirus expressing Dnmts or the empty plasmid vector, pCDH (*n* = 6, p<0.05, Student's *t*-test, mean ±s.e.m.).

DOI: https://doi.org/10.7554/eLife.30766.006

The following source data and figure supplements are available for figure 2:

**Source data 1.** Figure 2—source data.
DOI: https://doi.org/10.7554/eLife.30766.011
**Figure supplement 1.** Knock-down efficiency of Dnmt hairpins.
DOI: https://doi.org/10.7554/eLife.30766.007
**Figure supplement 1—source data 1.** Figure 2—figure supplement 1—source data.
DOI: https://doi.org/10.7554/eLife.30766.008
**Figure supplement 2.** Dnmt3a is necessary for Dex- and TNF-mediated IR.
DOI: https://doi.org/10.7554/eLife.30766.009
**Figure supplement 2—source data 1.** Figure 2—figure supplement 2—source data.
DOI: https://doi.org/10.7554/eLife.30766.010

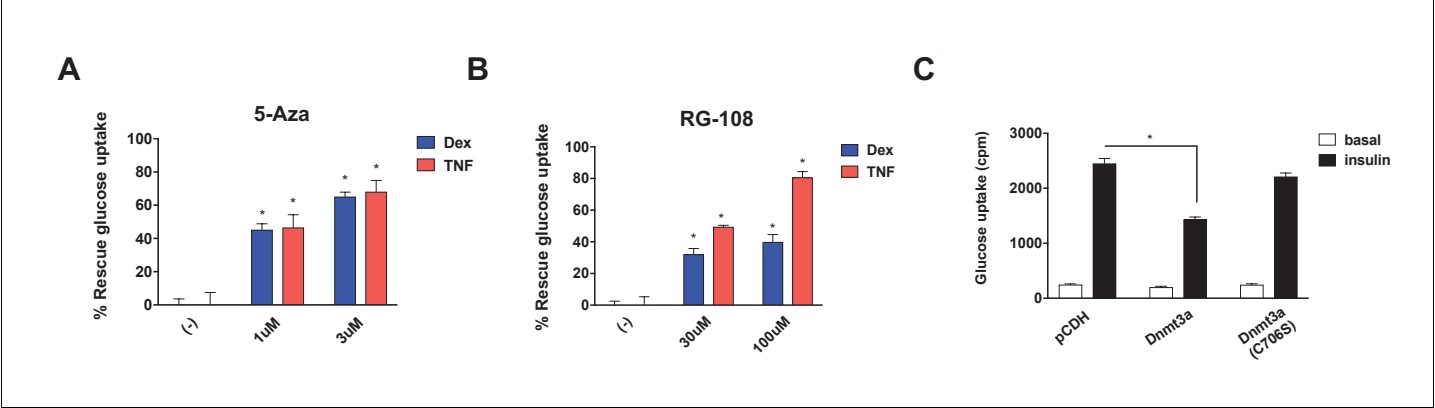

**Figure 3.** DNA methyltransferase activity is critical for Dnmt3a to mediate IR. Insulin-stimulated glucose uptake assay was conducted using 3T3-L1 adipocytes treated with Dnmt inhibitors (**A**, 5-Aza; **B**, RG-108) and Dex or TNF for 4 days. Shown is the % of insulin-stimulated glucose uptake rescued by Dnmt inhibitors (n = 6, p<0.05). (**C**) Basal and insulin-stimulated glucose uptake in mature 3T3-L1 adipocytes transduced with wild-type Dnmt3a vs. a catalytically inactive allele (C706S) (n = 6, p<0.05, Student's t-test, mean ±s.e.m.).

DOI: https://doi.org/10.7554/eLife.30766.012

The following source data and figure supplements are available for figure 3:

**Source data 1.** Figure 3—source data.
DOI: https://doi.org/10.7554/eLife.30766.015

**Figure supplement 1.** Dnmt inhibitors confer protection from Dex- and TNF-mediated IR.
DOI: https://doi.org/10.7554/eLife.30766.013

**Figure supplement 1—source data 1.** Figure 3—figure supplement 1—source data.
DOI: https://doi.org/10.7554/eLife.30766.014

*supplement 3A–C*). We next assessed whether insulin sensitization could be due to either enhanced browning or reduced inflammation of WAT. With regard to the former, we found no consistent change in the expression of major thermogenic genes such as *Ucp1* and *Ppargc1a* in either iWAT and BAT (*Figure 4—figure supplement 4A,B*). Additionally, gene expression analysis showed a non-significant trend toward reduced expression of proinflammatory genes such as *Tnfa* and *Ccl2* (which encodes MCP-1) in the eWAT of *Dnmt3a^AdiKO* mice (*Figure 4—figure supplement 5A*). We also saw no obvious change in the density of crown-like structures between WT and KO tissues (*Figure 4—figure supplement 5B*). Together, these results demonstrate that adipose-specific deletion of Dnmt3a confers protection from diet-induced IR in a manner that is independent of adiposity, inflammation, or browning of white adipose tissue.

## Unbiased gene profiling studies reveal that Fgf21 is a key metabolic target gene of Dnmt3a in adipocytes

Dnmt3a affects developmental processes by regulating non-overlapping sets of cell type-specific target genes (*Challen et al., 2011*; *Nguyen et al., 2007*; *Nishikawa et al., 2015*). To gain mechanistic insight into Dnmt3a-mediated IR at the molecular level, we performed RNA-seq in 3T3-L1 adipocytes, in both gain- and loss-of-function contexts. We have successfully used this approach to enhance our ability to identify *bona fide* targets (*Kang et al., 2012*). We found that 97 genes were repressed after shRNA-mediated Dnmt3a knock-down, while only eight genes were up-regulated in the presence of excess Dnmt3a; the single gene showing concordant regulation was Dnmt3a itself (*Figure 5A*). Conversely, 290 genes were induced by Dnmt3a knock-down, and 36 genes showed diminished expression after Dnmt3a overexpression; seven genes were concordantly regulated by both manipulations (*Figure 5B and C* and *Supplementary file 1*). These data are consistent with the expectation that Dnmt3a works primarily as a repressor of gene expression. Among the concordantly-regulated negative target genes of Dnmt3a, we were immediately struck by the presence of *Fgf21*. FGF21 is known to facilitate glucose uptake in adipocytes (*Ge et al., 2011*; *Lee et al., 2014*; *Minard et al., 2016*), and is a positive target gene of PPARγ (*Muise et al., 2008*). Furthermore, mice lacking FGF21 do not support Rosi-mediated insulin sensitization in vivo (*Dutchak et al., 2012*). We confirmed by Q-PCR that *Fgf21* expression is indeed reduced in cultured adipocytes



**Figure 4.** Adipose-specific Dnmt3a KO mice display improved whole body insulin sensitivity and glucose tolerance on HFD. (**A**) Body weight of *Dnmt3a*[AdiKO] and control mice on HFD (n = 7, 9, p<0.05, Student's *t*-test, mean ±s.e.m.). (**B**) Body composition after 10 weeks of HFD (n = 7, 9). (**C**) Effect of long-term fasting (16 hr) on body weight or body composition by EchoMRI. Shown is % loss after fasting (n = 14, 14, p<0.05, Student's *t*-test, mean ±s.e.m.). (**D**) Glucose tolerance test after 9 weeks on HFD (n = 7, 9, p<0.05, Student's *t*-test, mean ±s.e.m.). (**E**) The area under curve (AUC) in (**D**). (**F**) Insulin tolerance test after 11 weeks on HFD (n = 7, 9, p<0.05, Student's *t*-test, mean ±s.e.m.). (**G**) AUC in (**F**). (**H**) Fed and fasted insulin levels measured by ELISA (n = 7, 9 p<0.05, Student's *t*-test, mean ±s.e.m.) (**I**) Immunoblot of total and phospho-AKT and IRS-1 in eWAT after IP insulin. (**J**) Quantification of western blot in (**I**).

DOI: https://doi.org/10.7554/eLife.30766.016

The following source data and figure supplements are available for figure 4:

**Source data 1.** Figure 4—source data.

DOI: https://doi.org/10.7554/eLife.30766.027

**Figure supplement 1.** Adipose-specific deletion of Dnmt3a.

DOI: https://doi.org/10.7554/eLife.30766.017

**Figure supplement 1—source data 1.** Figure 4—figure supplement 1—source data.

DOI: https://doi.org/10.7554/eLife.30766.018

**Figure supplement 2.** Metabolic features of *Dnmt3a*[AdiKO] mice on chow.

DOI: https://doi.org/10.7554/eLife.30766.019

**Figure supplement 2—source data 1.** Figure 4—figure supplement 2—source data.

DOI: https://doi.org/10.7554/eLife.30766.020

**Figure supplement 3.** Metabolic features of female *Dnmt3a*[AdiKO] on HFD.

DOI: https://doi.org/10.7554/eLife.30766.021

**Figure supplement 3—source data 1.** Figure 4—figure supplement 3—source data.

DOI: https://doi.org/10.7554/eLife.30766.022

**Figure supplement 4.** Gene expression profile of thermogenic genes in *Dnmt3a*[AdiKO] mice on HFD.

DOI: https://doi.org/10.7554/eLife.30766.023

**Figure supplement 4—source data 1.** Figure 4—figure supplement 4—source data.

DOI: https://doi.org/10.7554/eLife.30766.024

*Figure 4 continued on next page*

*Figure 4 continued*

**Figure supplement 5.** Gene expression analysis of proinflammatory genes and histology examination of eWAT from *Dnmt3a^f/f* and *Dnmt3a^AdiKO* mice on HFD.
DOI: https://doi.org/10.7554/eLife.30766.025
**Figure supplement 5—source data 1.** Figure 4—figure supplement 5—source data.
DOI: https://doi.org/10.7554/eLife.30766.026

overexpressing Dnmt3a, is increased in cells expressing shDnmt3a, and most importantly, is up-regulated in the adipose tissue of *Dnmt3a^AdiKO* mice (*Figure 5D*). Unlike adipose tissue levels, serum levels of Fgf21 were reduced in HFD-fed *Dnmt3a^AdiKO* mice compared to controls (*Figure 5—figure supplement 5–*). To determine whether restoration of FGF21 levels could ameliorate Dnmt3a-mediated IR, we treated 3T3-L1 adipocytes that overexpress Dnmt3a with recombinant FGF21, which restored insulin sensitivity in a dose-dependent manner (*Figure 5E*).

## Dnmt3a-mediated changes in *Fgf21* gene expression involve changes in DNA methylation

In general, the level of DNA methylation at gene promoters is inversely correlated with mRNA expression (*Jones, 2012*). Given the functional significance of FGF21 in Dnmt3a-mediated IR, we further examined whether Dnmt3a directly alters methylation of the 5 kb upstream promoter regions of the *Fgf21* gene (*Figure 6A*). Using methylated DNA precipitation PCR (MeDIP-qPCR), we detected genomic DNA hypermethylation at *Fgf21* promoter regions in Dnmt3a overexpressor 3T3-L1 cells compared to controls (*Figure 6B*). Conversely, DNA hypomethylation of these same regions was seen in Dnmt3a knock-down cells (*Figure 6B*). In addition to the in vitro models, we conducted MeDIP-qPCR analysis using genomic DNA from fractionated adipocytes of *Dnmt3a^AdiKO* and control animals, finding reduced DNA methylation at the *Fgf21* promoter in KO cells (*Figure 6C*). Consistent with DNA methylation pattern at the Fgf21 promoter region, co-transfection of Dnmt3a potently inhibited promoter activity of *Fgf21* (*Figure 6D*). Together, these data suggest that Dnmt3a-mediated suppression of *Fgf21* involves changes in DNA methylation profile at key promoter regions.

## DNA methylation of *FGF21* is increased in human adipose tissue of subjects with type 2 diabetes

To translate some of our findings to humans, we investigated DNA methylation and gene expression of *FGF21* in human adipose tissue collected from diabetic vs. non-diabetic human subjects. In line with our mouse data, we found hypermethylation of four CpG sites annotated to *FGF21* in adipose tissue of subjects with type 2 diabetes compared with non-diabetic controls (*Figure 7A* and *Supplementary file 2*). Further, we found a negative correlation between DNA methylation and mRNA expression of *FGF21* in human adipose tissue (*Figure 7B–E*). Together, these data support a role for increased DNA methylation of *FGF21* in the regulation of adipose *FGF21* expression in association with metabolic disease.

## Discussion

While genetics plays an important role in obesity and T2D, genetic differences cannot fully explain many features of these conditions, such as discordance in monozygotic twins, and the close relationship with lifestyle factors (*McCarthy, 2010*; *Morris et al., 2012*; *Voight et al., 2010*; *Zeggini et al., 2008*). Of note, the vast majority of GWAS-proven allelic variants associated with T2D correspond much better to altered insulin secretion and islet function rather than to IR per se (*Voight et al., 2010*). Hence, forms of non-genetic variation, such as epigenetic alterations, must also be considered. This notion has been borne out by a recent epigenome-wide association study (EWAS) linking alterations in DNA methylation to whole-body insulin sensitivity (*Zhao et al., 2012*). Importantly, even genetic variants may cause disease in a DNA methylation-dependent manner, as has recently been shown for T2D and obesity (*Dayeh et al., 2013*; *Elliott et al., 2017*; *Nilsson et al., 2014*; *Olsson et al., 2014*; *Volkov et al., 2016*). We have also shown that adipose tissue from patients with T2D exhibit numerous methylation and expression differences relative to unrelated healthy controls, with many of the same changes also seen in twin pairs discordant for T2D (*Nilsson et al.,*



**Figure 5.** Fgf21 is a key target gene of Dnmt3a. Venn diagram showing the number of positive (A) and negative (B) Dnmt3a target genes through comparative analysis of RNA-seq profiles from Dnmt3a knockdown and overexpressing adipocytes. (C) Heat map showing differentially regulated genes by Dnmt3a overexpression and knock-down. Group 1: Genes that are down-regulated by Dnmt3a overexpression; Group 2: Genes that are down-regulated by Dnmt3a overexpression and up-regulated by Dnmt3a knock-down; Group 3: Genes that are up-regulated by Dnmt3a knock-down. (D) *Fgf21* mRNA expression in Dnmt3a overexpressor (*left*) and knock-down (*middle*) L1 adipocytes and in fractionated adipocytes (*right*) from *Dnmt3a[AdiKO]* and *Dnmt3a[f/f]* mice on HFD (n = 5). (E) Mature 3T3-L1 adipocytes expressing GFP or Dnmt3a treated with indicated amount of recombinant Fgf21 for 48 hr (or vehicle), and tested for insulin sensitivity by $^{3}$H-2-DG assay. Shown is basal and insulin-stimulated glucose uptake. (*n* = 6 mice, p<0.05, Student's *t*-test, mean ±s.e.m.).

DOI: https://doi.org/10.7554/eLife.30766.028

The following source data and figure supplements are available for figure 5:

**Source data 1.** Figure 5—source data.
DOI: https://doi.org/10.7554/eLife.30766.031
**Figure supplement 1.** Serum Fgf21 levels in *Dnmt3a[AdiKO]* mice.
DOI: https://doi.org/10.7554/eLife.30766.029
**Figure supplement 1—source data 1.** Figure 5—figure supplement 1—source data.
DOI: https://doi.org/10.7554/eLife.30766.030

*2014*). In this study, we identified increased expression of an enzymatic effector of DNA methylation, Dnmt3a, as a key epigenetic driver of IR in vitro and in vivo.

DNA methylation, like other forms of epigenomic modification, has been an attractive therapeutic target because of their plasticity and because they offer an opportunity to reprogram cells into a more healthy state. For example, administration of pan-inhibitors of histone deacetylase exerts beneficial metabolic effects in both mice and humans, such as increased energy expenditure, insulin sensitivity and secretion (*Christensen et al., 2011*; *Daneshpajooh et al., 2017*; *Sharma and Taliyan, 2016*; *Ye, 2013*). DNA methyltransferase inhibitors have also been shown to have an insulin-

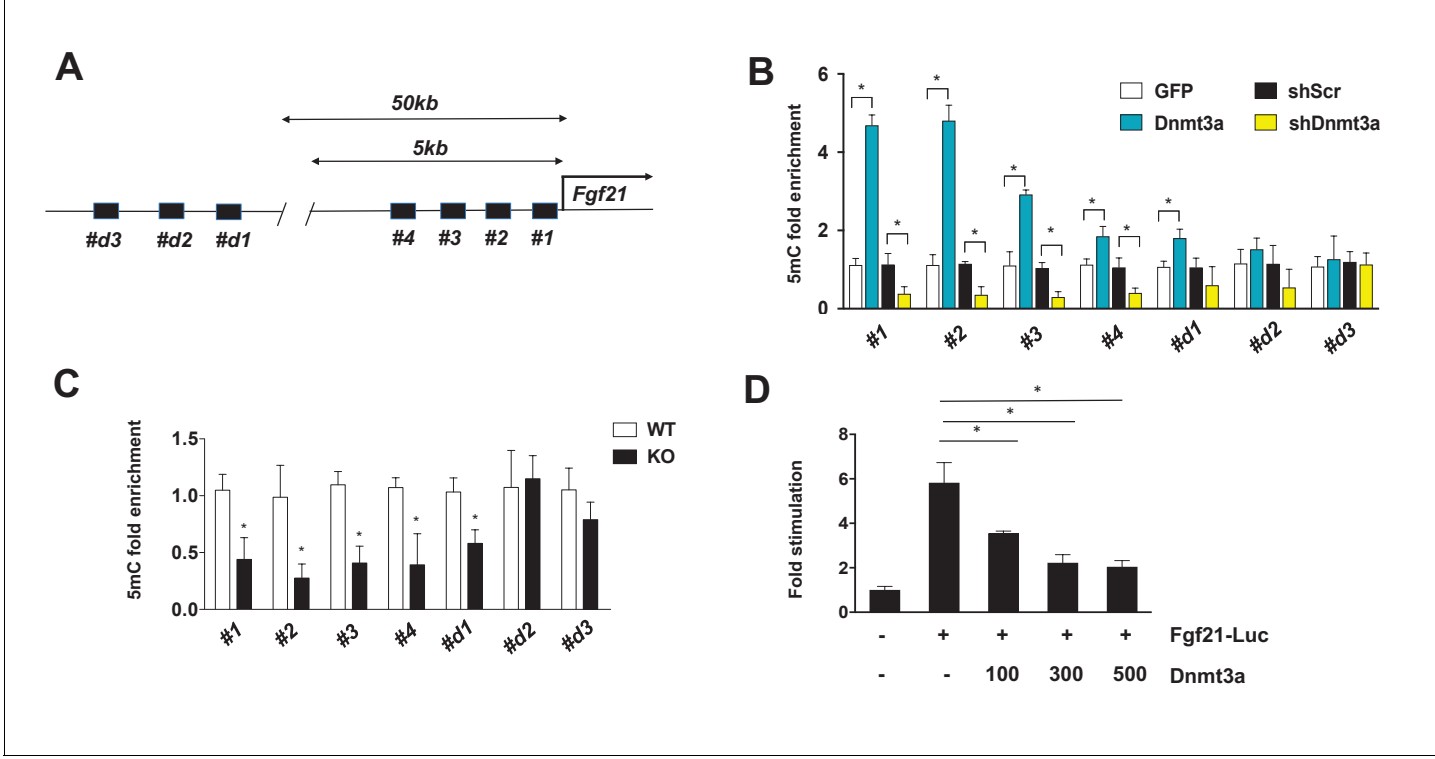

**Figure 6.** Dnmt3a-mediated methylation of the *Fgf21* gene promoter. (**A**) Schematic showing the tested regions for MeDIP-qPCR at the *Fgf21* locus. Target regions were chosen using MethFinder (http://www.urogene.org/methprimer) on regions proximal (5 kb) and distal (50 kb) to the *Fgf21* transcription start site. (**B**) MeDIP-qPCR was performed using genomic DNA extracted from 3T3-L1 adipocytes transduced with GFP vs. Dnmt3a or shScr vs. shDnmt3a. Shown is relative fold enrichment of 5mC of Dnmt3a overexpressor and KD cells over control cells. (**C**) MeDIP-qPCR was performed using genomic DNA from Dnmt3a KO and Flox mice on HFD. Shown is relative fold enrichment of 5mC of KO over WT samples. (**D**) Fgf21-promoter reporter (1497/+5) assay performed by co-transfection of the indicated amounts of Dnmt3a expression vector. (*n* = 3 mice, p<0.05, Student's *t*-test, mean ±s.e.m.).

DOI: https://doi.org/10.7554/eLife.30766.032

The following source data is available for figure 6:

**Source data 1.** Figure 6—source data.
DOI: https://doi.org/10.7554/eLife.30766.033

sensitizing effect in vivo and in vitro (**Kim et al., 2015**). In line with pharmacological studies, mutant mouse models carrying loss-of function histone modifiers (e.g. *Mll2, Ehmt1, Jmhd2a Lsd1*) were shown to have profound impact on whole body metabolism (**Lee et al., 2006**; **Ohno et al., 2013**; **Sambeat et al., 2016**; **Tateishi et al., 2009**; **Zeng et al., 2016**). It should be noted that metabolic consequences in mouse models carrying mutations of histone modifiers appear to be mainly attributable to changes in body weight. Unlike these examples, adipose-specific Dnmt3a KO mice are relatively unique in that they display improved insulin sensitivity without a change in adiposity or body weight. Similarly, the effect on insulin action is not dependent on increased beige or brown adipocytes, or on reducing intra-adipose inflammation.

Our in vitro study indicates that overexpression of Dnmt3a is sufficient to induce cell autonomous IR, however, it does not seem to be the case in vivo. Kamei et al showed that transgenic expression of Dnmt3a per se was not sufficient to render whole body IR and glucose intolerance (**Kamei et al., 2010**) on chow, HFD, or high methyl diet with chow diet. Authors observed only a mild increase in the gene expression of proinflammatory cytokines (i.e. *Mcp1* and *Tnfa*) when animals were put on HFD. It should be noted that the major caveat of this study is the use of aP2 promoter to drive the transgene expression (**Jeffery et al., 2014**; **Kang et al., 2014**; **Lee et al., 2013**); Potentially, non-adipose expression and earlier expression during development might have a compounding factor in this study. Despite these notions, it will be highly interesting to address how the combination of



**Figure 7.** *FGF21* DNA methylation in adipose tissue from subjects with T2D compared with controls. (**A**) Four CpG sites annotated to *FGF21* with significant difference in adipose tissue DNA methylation between subjects with type 2 diabetes and controls. ($p<0.01$, paired Wilcoxon statistics, mean ±s.d.). (**B–D**) Correlations between DNA methylation and mRNA expression for CpG sites significantly associated with T2D in human adipose tissue. (r = Pearson correlation coefficient.).

DOI: https://doi.org/10.7554/eLife.30766.034

high methyl donor diet and HFD would affect the metabolic profile of these transgenic animals compared to controls.

Our unbiased gene expression profiling studies led us to identify *Fgf21* as a key target gene in adipocytes, and the data from diabetic humans suggests that this could be a conserved mechanism of IR in both mice and humans. Fgf21 is better known as a hepatokine, and the liver is the main source of circulating Fgf21 (*Markan et al., 2014*) Adipose Fgf21 probably does not circulate, but rather acts in an autocrine/paracrine manner (*Dutchak et al., 2012*; *Markan et al., 2014*). In fact, levels of Fgf21 in serum were moderately reduced in Dnmt3a KO animals compared to controls, in contrast to significantly elevated adipose tissue levels. Studies have shown a positive correlation between insulin and circulating Fgf21 levels (*Chavez et al., 2009*; *Chen et al., 2008*; *Zhang et al., 2008*), suggesting that insulin may be a key determinant of Fgf21 in serum. Thus, it is possible that the reduced insulin levels in Dnmt3a KO mice on HFD may have led to reduction of Fgf21 production in liver. This notion is in line with the finding that Fgf21 serum levels are often elevated in obesity-associated IR (*Fisher and Maratos-Flier, 2016*; *Morrice et al., 2017*), possibly reflecting resistance to Fgf21 (*Fisher et al., 2010*).

We propose that the local activity of Fgf21 within the fat pad is important for the improved insulin sensitivity in the knock-out mice, although formal proof of this idea would require generating adipose tissue-specific Dnmt3a and Fgf21 double knockout mice. In line with this view, there are previous studies indicating that adipose Fgf21 is insulin-sensitizing. For example, three independent studies demonstrated that mutant mice devoid of Fgf21 signaling in adipose tissue secondary to ablation of FGFR1 or bKlotho, were refractory to the insulin-sensitizing effect of Fgf21 administration

(*Adams et al., 2013*; *BonDurant et al., 2017*; *Ding et al., 2012*). Some of these studies (*Adams et al., 2013*; *Ding et al., 2012*) suggested that adiponectin is an essential mediator of Fgf21 metabolic action; the other study (*BonDurant et al., 2017*) suggested that adiponectin is dispensable. In our studies, albeit it did not reach statistical significance, there was a trend toward increased mRNA expression of *Adipoq* in both L1 and primary Dnmt3a KO adipocytes (not shown).

Control of Fgf21 expression appears to be highly tissue-specific. For example, PPARγ is an important driver of Fgf21 in adipose tissue, while PPARα serves that role in liver (*Dutchak et al., 2012*). Other studies have also demonstrated that Fgf21 is positively regulated by PPARγ in fat (*Muise et al., 2008*; *Wang et al., 2008*). In addition to focused studies on *Fgf21* promoter regions, global DNA methylation profiling will be necessary to identify differentially methylated regions in the setting of Dnmt3a gain- and loss-of function.

A recent study demonstrated that Dnmt1 may contribute to obesity-associated inflammation and insulin resistance by promoting hypermethylation at the *Adipoq* locus (*Kim et al., 2015*). Concordant with this study, we also see elevated Dnmt1 levels in the adipose tissue of obese animals. However, knockdown of Dnmt1 did not rescue insulin sensitivity in the setting of Dex or TNF, and overexpression of Dnmt1 did not reduce insulin-stimulated glucose uptake in cultured adipocytes. Based on those criteria, we chose to focus on Dnmt3a, and showed that adipose-specific deletion of this gene promoted insulin sensitivity even in weight-matched obese animals. Of note, we did not identify *Adipoq* as a regulated target after either deletion or overexpression of Dnmt3a.

Despite some effort, we have not yet identified the mechanism by which Dnmt3a expression is increased in obesity. We found that Dnmt3a is not induced by lipids (e.g. palmitate), pro-inflammatory signals (e.g. IL-6, LPS), high doses of insulin, or high glucose in cultured adipocytes (not shown). In our study, in vivo administration of thiazolidinediones normalized the expression of Dnmt3a in *ob/ob* mice but not in lean control animals. The mechanism by which PPARγ represses gene expression is not entirely clear. Trans-repression through binding of inflammatory transcription factors like NF-κB and AP-1 has been proposed as a general mechanism for this type of effect (*Glass and Saijo, 2010*) although we note that there are PPARγ binding sites near the *Dnmt3a* TSS (not shown). It is thus unclear if direct or indirect (i.e. trans-repression) mechanisms account for the actions of rosiglitazone on *Dnmt3a* expression in obesity.

In summary, we have identified a novel role for Dnmt3a as an epigenetic mediator of adipose IR in vitro and in vivo. Future studies will be required to fully elucidate the full range of its target genomic regions and metabolic effects.

## Materials and methods

### Cell culture

3T3-L1 preadipocytes were obtained from ATCC and maintained and differentiated as described (*Kang et al., 2012*). These cells were authenticated by the ability to differentiate and confirmed to be mycoplasma negative. To generate lentivirus particles, lentiviral constructs were co-transfected with pM2DG- and psPAX-expressing plasmids into 293T cells. After 48 hr, virus-containing supernatant was collected, filtered through 0.45 μm filters, and added to mature 3T3-L1 adipocytes for 24 hr along with 8 μg/ml Polybrene. Transduction efficiency was determined by comparison with cells transduced in parallel with a GFP-expressing lentivirus. For the ex vivo system, subcutaneous adipose tissue from wild-type C57Bl/6 mice was fractionated with digestion buffer (10 mg/ml collagenase D, 2.4 units of dispase II, 10 mM $CaCl_2$ in PBS). Cells from the stromal-vascular fraction (SVF) were plated in culture and differentiated as described previously (*Qureshi et al., 2014*). To differentiate hASCs, Rosi (5 μM) was added in addition to IBMX (0.5 mM), insulin (5 μg/ml), and dexamethasone (1 μM), from day 0–2.

### ³H-2-DG assay

Mature 3T3-L1 adipocytes were incubated in serum-free DMEM for 4–6 hr. Cells were then washed three times with KRH buffer (12 mM HEPES, pH 7.4, 121 mM NaCl, 5 mM KCl, 0.33 mM $CaCl_2$, and 1.2 mM $MgSO_4$) and incubated for 20 min in KRH buffer in the absence or presence of 50 nM insulin. Cells were treated with 2-deoxy-d-[2,6-³H]-glucose (0.33 μCi/ml) for another 10 min. Glucose uptake was stopped quickly by three rapid washes with KRH buffer containing 200 mM glucose and 10 μM

cytochalasin B on ice. Cells were solubilized in 0.1% SDS for 30 min, and radioactivity was measured by liquid scintillation counting.

## Reagents

2-Deoxy-d-[2,6-³H]-glucose was purchased from PerkinElmer NEN radiochemicals. Insulin, dexamethasone, isobutylmethylxanthine (IBMX), cytochalasin B, glucose, 2-deoxyglucose, 5-Aza, ECCG, and TF-3 were purchased from Sigma-Aldrich. Recombinant TNF-α and Fgf21 were purchased from Millipore. RG-108 was purchased from Stemgent.

## Antibodies

Antibodies were purchased from Cell Signaling (Dnmt3a, 3598S; Akt, 9272; pAkt [S473], 3787; IRS-1, 2382), and from Thermo Fisher (β-actin, MA5-14739, pIRS1[pY612], 44–816G).

## Animals

For rosiglitazone studies, female mice *ob/ob* and *ob/+*were treated starting at 8 weeks of age with rosiglitazone by gavage (10 mg/kg body weight) in 0.5% carboxymethylcellulose versus 0.5% carboxymethylcellulose vehicle control daily for 6 weeks. For chow and high-fat feeding studies, male C57Bl/6J mice were put on diet beginning at 8 weeks of age and continued for three months ($n = 8$ per dietary condition). Samples were collected from the perigonadal fat pad.

For histology, adipose tissues were fixed with neutral-buffered formalin and embedded in paraffin and sections were stained with H&E.

For in vivo insulin signaling assay, after overnight fast, insulin (10 U/kg, 5 min) or saline IP were given to WT and KO mice on HFD. After 10 min various tissues were harvested and stored at –80°C until use. Tissue samples were homogenized in cell signaling lysis buffer containing protease inhibitors (Roche) and phosphatase inhibitors (Sigma-Aldrich) and subjected to western blotting. All animal work was approved by the BIDMC IACUC and/or the UC Berkeley ACUC.

## RNA extraction and quantitative PCR

Total RNA was extracted from cells or tissues using TRIzol reagent according to the manufacturer's instructions. cDNA was reverse-transcribed from 1 μg of RNA using the RETROscript first strand synthesis kit (Ambion). Quantitative PCR (qPCR) was performed with SYBR Green qPCR Master Mix (Applied Biosystems) using a 7900HT Fast Real-Time PCR System (Applied Biosystems) and CFX96 Touch (Bio Rad). Primer sequences are listed in *Supplementary file 3*. The relative amount of mRNA normalized to cyclophilin B was calculated using the delta–delta method (*Livak and Schmittgen, 2001*).

## Plasmids

Hairpins against Dnmt1, Dnmt3a, and Dnmt3b were purchased from Sigma. Lentiviral overexpression vectors for Dnmt1, 3a, and 3b were subcloned into pCDH using various multicloning sites (XbaI/NotI for Dnmt1, EcoRI/NotI for Dnmt1, Dnmt3a, Dnmt3a-CM) and NotI sites and hairpins targeting *Dnmts* were subcloned at AgeI/EcoRI or purchased from Open Biosystems. Hairpin sequences are shown in *Supplementary file 3*. Fgf21-luc (1497/+5) was generously provided by Dr. Steven Kliewer (UT Southwestern).

## MeDIP-qPCR

Genomic DNA was sheared by sonication to an average of 200–800 bp size. Two microgram of denatured DNA was incubated with 2 μg of anti-5-methylcytidine antibody (Epigentek) in IP buffer (10 mM Na-Phosphate pH 7.0, 0.14 M NaCl, 0.05% Triton X-100) for 2 hr at 4°C. Antibody-bound DNA was collected with 20 μl of Dynabeads anti-mouse IgG (Invitrogen Dynal, Oslo, Norway) for 1 hr at 4°C on a rotating wheel and successively washed five times with washing buffer (0.1% SDS, 1% Triton X-100, 2 mM EDTA, 20 mM Tris-HCl pH 8.1, 150 mM NaCl), and twice with TE (10 mM Tris·Cl, 1 mM EDTA pH 8.0). DNA was recovered in digestion buffer 125 μl buffer (50 mM Tris pH 8.0, 10 mM EDTA, 0.5% SDS, 35 μg proteinase K) and incubated for 3 hr at 65°C. Recovered DNA was used for qPCR analysis. Primers for MeDIP-qPCR studies are listed in *Supplementary file 3*. All data are normalized to input.

## RNA-seq studies

mRNA was purified from 100 ng of total RNA using the Ribo-Zero Magnetic Gold Kit (catalog MRZG126, Illumina). Libraries were prepared using the TruSeq RNA Library Preparation Kit v2 (catalog RS-122–2001, Illumina) according to the manufacturer's protocol starting with the EPF step. Sequencing was performed on the Illumina HiSeq2500. RNA-Seq data are aligned using TopHat2 (*Kim et al., 2013*). Reads are assigned to transcripts using featureCounts and an mm9 genome modified to minimize overlapping transcripts (*Liao et al., 2014*). Differential expression analysis of the data is performed using *EdgeR* (*Robinson et al., 2010*).

## DNA methylation and mRNA expression in human adipose tissue

DNA methylation was analyzed in adipose tissue from 28 subjects with type 2 diabetes and 28 controls as well as in adipose tissue of 14 monozygotic twin pairs (n = 28) using Infinium HumanMethylation450 BeadChips (Illumina) according to our previous study where we also present the characteristics of these subjects (*Nilsson et al., 2014*). Their characteristics are presented in *Supplementary file 4*. mRNA was analyzed in adipose tissue from the 12 monozygotic twin pairs (n = 24) using GeneChip Human Gene 1.0 ST arrays (Affymetrix, Santa Clara, CA) according to the manufacturer's recommendations. In the present study, we studied mRNA expression and DNA methylation of CpG sites annotated to *FGF21*. DNA and RNA were extracted from human adipose tissue as presented (*Nilsson et al., 2014*).

## Acknowledgements

Work was funded by NIH R01 102173, 102170, and 085171 to EDR and AHA Award #15SDG25240017 to SK. We thank Jeff Leung for technical assistance and the Rosen and the Kang Lab members for helpful conversations. We are also grateful to Drs. Hei Sook Sul, Jen-Chywan Wally Wang, and Andreas Stahl for constructive comments.

## Additional information

### Funding

| Funder | Grant reference number | Author |
|---|---|---|
| National Institutes of Health | 102173 | Evan D Rosen |
| National Institutes of Health | 102170 | Evan D Rosen |
| National Institutes of Health | 085171 | Evan D Rosen |
| American Heart Association | 15SDG25240017 | Sona Kang |
| National Institutes of Health | 116008 | Sona Kang |

The funders had no role in study design, data collection and interpretation, or the decision to submit the work for publication.

### Author contributions

Dongjoo You, Formal analysis, Validation, Investigation, Carried out most of biological experiments and assisted with preparing the manuscript; Emma Nilsson, Data curation, Formal analysis, Investigation, Collected human adipose tissue and analyzed DNA methylation and gene expression with human sample; Danielle E Tenen, Investigation, Methodology, Generated RNA-Seq library; Anna Lyubetskaya, Formal analysis, Generated and analyzed RNA-Seq data; James C Lo, Resources, Helped with metabolic characterization of knock-out mouse models; Rencong Jiang, Investigation, Carried out insulin signaling assay; Jasmine Deng, Investigation, Carried out gene expression analysis of several biological experiments; Brian A Dawes, Formal analysis, Helped with RNA-Seq data analysis, Generated figures, Submitted GEO data; Allan Vaag, Investigation, Collected human adipose tissue and analyzed DNA methylation and gene expression with human sample; Charlotte Ling, Resources, Investigation, Collected human adipose tissue and analyzed DNA methylation and gene expression data, Supervised overall human data set; Evan D Rosen, Supervision, Funding acquisition,

Writing—original draft, Writing—review and editing, Supervised experiments and edited the manuscript and revision for the revision; Sona Kang, Supervision, Investigation, Writing—original draft, Writing—review and editing, Designed most of biological experiments, Supervised experiments, Wrote draft and revision for the revision

### Author ORCIDs
Sona Kang https://orcid.org/0000-0002-9831-677X

### Ethics
Animal experimentation: This study was performed in strict accordance with the recommendations in the Guide for the Care and Use of Laboratory Animals of the National Institutes of Health. All animal work was approved by the BIDMC IACUC (056-2017) and/or the UC Berkeley ACUC (AUP-2015-08-7887).

### Decision letter and Author response
Decision letter https://doi.org/10.7554/eLife.30766.041
Author response https://doi.org/10.7554/eLife.30766.042

## Additional files

### Supplementary files
• Supplementary file 1. Genes that are up-regulated (UP) and down-regulated (DOWN) Dnmt3a vs. GFP and shDnmt3a/shScr.
DOI: https://doi.org/10.7554/eLife.30766.035

• Supplementary file 2. DNA methylation of four CpG sites annotated to Fgf21 showing differential DNA methylation in adipose tissue of a case control study for type 2 diabetes.
DOI: https://doi.org/10.7554/eLife.30766.036

• Supplementary file 3. Oligonucleotide sequences used in this manuscript.
DOI: https://doi.org/10.7554/eLife.30766.037

• Supplementary file 4. Clinical characteristics of study subjects included in the discordant twin cohort and the case control cohort.
DOI: https://doi.org/10.7554/eLife.30766.038

• Transparent reporting form
DOI: https://doi.org/10.7554/eLife.30766.039

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
