## [Decision Letter]

Thank you for submitting your article "Dnmt3a is an epigenetic mediator of adipose insulin resistance" for consideration by *eLife*. Your article has been favorably evaluated by Mark McCarthy (Senior Editor) and three reviewers, one of whom, Clifford J Rosen (Reviewer #1), is a member of our Board of Reviewing Editors. The following individual involved in review of your submission has agreed to reveal their identity: Rob Koza (Reviewer #2).

The reviewers have discussed the reviews with one another and the Reviewing Editor has drafted this decision to help you prepare a revised submission. Overall there is significant enthusiasm for this manuscript, if the suggestions noted below by the reviewing editor are considered within the context of the other comments and the time frame for resubmission.

Essential revisions:

There were a couple of points that need to be addressed in the response and incorporated within the manuscript. These include three major comments:

1) A clearer rationale for studying epididymal fat as the major tissue source for the studies and elaboration about the possible beiging that could occur in the conditional deletion as a mechanism for the enhanced insulin sensitivity.

2) More elaboration about the regulation of the enzyme itself and the impact of activated PPARG on expression of Dnmt3.

3) Considerations for more in vivo proof that FGF-21 is the target repressor of Dnmt3.

*Reviewer #1:*

Rosen and colleagues in this paper found enhanced expression of DNA methyltransferases (Dnmts) in insulin resistance. Pharmacological and genetic inhibition studies demonstrated that Dnmt3a was both necessary and sufficient to mediate insulin resistance in cultured mouse and human adipocytes. In addition, adipose-specific Dnmt3a knock-out mice were protected from diet-induced insulin resistance and glucose intolerance without accompanying changes in adiposity. Gene expression profiling studies revealed Fgf21 as a key negatively regulated Dnmt3a target gene in adipocytes with concordant changes in DNA methylation at the Fgf21 promoter region. Fgf21 rescued Dnmt3a-mediated insulin resistance in vitro, and DNA methylation at the FGF21 locus was elevated in human subjects with diabetes and correlated negatively with expression of FGF21 in human adipose tissue. Thus they concluded that adipose Dnmt3a is a novel epigenetic mediator of insulin resistance in vitro and in vivo.

This manuscript breaks new ground in describing an epigenetic mechanism for insulin resistance that is mediated through adipose specific FGF-21. The experiments are straightforward, the results consistent and the discussion is not over-reaching. The in vivo data are strong, and the in vitro rescue and silencing studies stand up to closer scrutiny. The human studies provide a translational perspective although somewhat limited by the confounders in T2d. The dose dependent suppression of Fgf21 luc with Dnmt3a in Figure 6 is particularly striking. Thus overall there are considerable strengths that are both basic and translational. The concerns are relatively limited and are noted below:

1) Rosiglitazone suppressed Dnmt3a – have the authors looked at whether activated PPARG directly mediates the suppression of Dnmt3a expression, or is it one of the many downstream targets of PPARG? This is particularly relevant since the enzyme's regulation is not defined, although it increases with a HFD, albeit without a known mechanism.

2) Can the authors discuss the discrepancy in vivo between circulating FGF-21 (likely hepatic- low) and adipose FGF-21 levels (high) in the Dnmt3a KO mice?

3) The authors mention that Fgf21 null mice do not support Rosi-mediated insulin sensitivity – does crossing the Fgf21-/- with the Dnmt3 adipose specific KO restore insulin resistance in those mice fed a HFD? This certainly would provide further in vivo proof of concept that FGF-21 is the target repressor of Dnmt3a.

4) The human studies are interesting but somewhat limited by the small number of subjects and the relatively weak, albeit significant negative regressions between methylation at the Fgf21 locus and mRNA expression at CpG sites. Of concern is the lack of a description of the diabetic subjects and their controls. For example, were the T2D patients treated with oral agents and/or insulin and did concurrent treatment affect methylation expression status?

5) One of the novel aspects of these studies is the similar weight gain and fat mass change in the Drnmt3a adipose specific KO, compared to the control. Thus insulin sensitivity is maintained even in the face of greater adiposity, a key finding that needs more emphasis in the Results and Discussion.

6) The authors fail to cite the paper by Kamei et al. in Obesity 2010, which demonstrated that an aP2 transgenic Dnmt3 mouse (PMID: 19680236) exhibit similar weight gain to wt mice; but importantly those mice have increased expression of Tnfa and MCP-1 in fat from their adipose depots. Similarly in that paper Dnmt3 expression was higher in WT mice fed a HFD. There should be a citation and also discussion about that paper (limitations based on the aP2 promoter, etc.).

*Reviewer #2:*

The manuscript entitled: 'Dnmt3a is an epigenetic mediator of adipose insulin resistance' submitted presents substantial and compelling evidence that adipose tissue Dnmt3a plays a regulatory role in insulin resistance. The PIs present a clear and logical progression of studies with knockdown and transgenic overexpression of Dnmt3a in cultured adipocytes; and, adipose tissue specific inactivation of Dnmt3a in mice. In addition, RNA-seq analyses of 3T3-L1 adipocytes with shRNA knockdown or transgenic overexpression of Dnmt3a identified Fgf21 as a Dnmt3a target gene. Analyses of the Fgf21 promoter using MeDIP-qPCR confirmed DNMT3A-mediated methylation as a mechanism for Fgf21 repression. Overall, these studies are comprehensive and very well presented; and, begin to elucidate epigenetic mechanisms that can modulate adipose tissue 'health' with respect to insulin resistance and glucose uptake.

1) What is the rationale for using the epididymal adipose tissue depot to measure HFD-mediated induction of Dnmt3a? Are there differences in HFD regulation of Dnmt3a between visceral and subcutaneous adipose tissue? Fgf21?

2) Although not as dramatic as in adipose tissue, there is a relatively robust and significant increase of Dnmt3a in skeletal muscle in mice fed a HFD (Figure 1—figure supplement 1). What is the relative expression of skeletal muscle Dnmt3a with respect to adipose tissue and liver? What are the potential implications for HFD-mediated up-regulation of skeletal muscle Dnmt3a with respect to Fgf21 expression?

*Reviewer #3:*

The manuscript "Dnmt3a is an epigenetic mediator of adipose insulin resistance" by You et al. presents new data linking the DNA methyltransferase Dnmt3a to insulin resistance in adipocytes. The authors start by demonstrating increased expression of Dnmt3a in models of insulin resistance. They then do pharmacologic and genetic studies in vitro to support a role for Dnmt3a in adipocyte insulin sensitivity. They then go on to generate an adipocyte-specific Dnmt3a knockout mouse and show that it is protected from diet-induced insulin resistance. Finally, they perform mechanistic studies showing that Fgf21 is negatively regulated by Dnmt3a. This is a well-written paper, with novel data, and well-designed experiments. As such, this would be of broad interest to the readership of this journal.

If the authors are able to address the point below, the manuscript would be further strengthened:

1) In their studies here, the authors only look at Dnmt3a in gonadal white fat (eWAT). Since Dnmt3a is apparently broadly expressed, it would be interesting to know whether it is also increased in subcutaneous/beige fat and in brown fat in models of insulin resistance. Moreover, it would be relevant to know whether beige/brown fat are altered in the adipocyte-specific Dnmt3a knockout model. If for example, the mutant mouse has activated brown or beige fat, this could be an explanation for the improved glucose and insulin tolerance.

---

## [Author Response]

Essential revisions:There were a couple of points that need to be addressed in the response and incorporated within the manuscript. These include three major comments:1) A clearer rationale for studying epididymal fat as the major tissue source for the studies and elaboration about the possible beiging that could occur in the conditional deletion as a mechanism for the enhanced insulin sensitivity.

We now show that visceral fat depots (i.e. epididymal and mesenteric WAT) express higher levels of Dnmt3a than liver and muscle, and we also show that the levels in these depots are more responsive to the effects of HFD than inguinal WAT and interscapular BAT. Additionally, we have looked for evidence of subcutaneous browning but we see none, either by histological examination or by analysis of thermogenic gene expression.

2) More elaboration about the regulation of the enzyme itself and the impact of activated PPARG on expression of Dnmt3.

While we agree that this is an interesting topic, it is very difficult to tease this apart, as explained below in the response to reviewer 1. We have done several things to address this issue that have produced data but not a tremendous amount of insight. To summarize, we have tried to model the effects of obesity in cultured adipocytes using FFAs, cytokines, insulin, and glucose, none of which raise Dnmt3a levels. Treatment of cultured adipocytes with TZD does not reduce basal levels of Dnmt3a, which is consistent with the in vivo results showing that Rosi reduces Dnmt3a levels in obese but not lean mice. There are PPARγ binding sites in the murine genome that lie near the Dnmt3a TSS; whether these are functional is not easy to test given the constraints already mentioned. Complicating this further is the fact that inhibiting PPARγ in mature adipocytes leads to a general process of de-differentiation, confounding efforts to parse a specific effect on Dnmt3a expression. Very little is known about how PPARγ (or any nuclear receptor) mediates negative gene expression; we submit that detailed analysis of this is beyond the scope of this study.

3) Considerations for more in vivo proof that FGF-21 is the target repressor of Dnmt3.

The definitive experiment to demonstrate this in vivo would be a five allele cross of adiponectinCre, Dnmt3a Flox, and Fgf21 null mice. This experiment can’t be completed within the time frame laid out by the *eLife* review process, and completing it would require at least a year and half of breeding. We have therefore chosen not to pursue this as part of the current manuscript.

Reviewer #1:[…] 1) Rosiglitazone suppressed Dnmt3a – have the authors looked at whether activated PPARG directly mediates the suppression of Dnmt3a expression, or is it one of the many downstream targets of PPARG? This is particularly relevant since the enzyme's regulation is not defined, although it increases with a HFD, albeit without a known mechanism.

We appreciate (and share) the reviewer’s curiosity about this important issue, but we would argue that is a very complex subject to address. First, we know very little about negative regulation by PPARγ in general. Like many nuclear receptors that are primarily transcriptional activators, PPARγ has negative targets, including leptin and adipsin. How this occurs is unknown, although trans-repression has been proposed as a mechanism for TZD antagonism of inflammatory genes. We note that there are bona fide PPARγ sites near the *Dnmt3a* locus (see Author response image 1), suggesting at least the possibility of direct regulation. Proving that would require CRISPRing those sites in mature adipocytes; not an easy feat. One can’t simply perform loss of function studies of PPARγ in mature adipocytes, as the cells immediately begin to dedifferentiate. This has an effect on Dnmt3a gene expression, but not the effect we are looking to study. Given these constraints, we did the following: first, we assessed whether lipids, glucose, insulin, or cytokine administration could model the effect of obesity on Dnmt3a expression in vitro. Unfortunately, they do not. Next, we tested whether rosiglitazone could repress Dnmt3a in 3T3-L1 adipocytes, and if so, whether cycloheximide would have an effect on the process. Those data are shown below. The upshot is that TZD does not repress basal Dnmt3a in adipocytes, consistent with the in vivo data shown in Figure 1, in that TZD rescues the increased Dnmt3a of obesity but does not cause Dnmt3a levels to go lower than those seen in lean animals. We have added discussion of these points to the manuscript. We have not included the data shown in Author response image 1 because we feel they don’t bring additional enlightenment to the reader, but we can do so if the reviewer and editor wish.

**Author response image 1. respfig1:** (**A**) Histograms of PPARγ-ChIP and histone modification fragments near the mouse *Dnmt3a* locus (Mikkelsen et al., 2010). Depicted with arrow are PPARγ peaks that co-localize with H3K4me1 and/or H3K27ac enhancer peaks. (**B**) The effect of Rosi (1uM) on the expression of Dnmt3a in the presence/absence of cycloheximide (10ug/ml) in mature 3T3-L1 adipocytes.

2) Can the authors discuss the discrepancy in vivo between circulating FGF-21 (likely hepatic- low) and adipose FGF-21 levels (high) in the Dnmt3a KO mice?

We have added additional thoughts on this topic to the Discussion.

3) The authors mention that Fgf21 null mice do not support Rosi-mediated insulin sensitivity – does crossing the Fgf21-/- with the Dnmt3 adipose specific KO restore insulin resistance in those mice fed a HFD? This certainly would provide further in vivo proof of concept that FGF-21 is the target repressor of Dnmt3a.

We absolutely agree that this is the definitive in vivo experiment. However, it is a five allele cross (two Dnmt3a^flox^ alleles, two FGF21 KO alleles, and one adiponectin Cre), and we fear that the time spent on this (approximately 18 months) would unnecessarily delay this report and would dilute the impact of our findings.

4) The human studies are interesting but somewhat limited by the small number of subjects and the relatively weak, albeit significant negative regressions between methylation at the Fgf21 locus and mRNA expression at CpG sites. Of concern is the lack of a description of the diabetic subjects and their controls. For example, were the T2D patients treated with oral agents and/or insulin and did concurrent treatment affect methylation expression status?

Based on this valid comment, we added a table (see Supplementary file 4) describing the characteristics of the diabetics and the controls included in this study. We also added information about the treatments of the diabetics in the legend of the table. We did not find any effects of treatment (insulin and metformin) on methylation of three of the studied CpG sites or expression of *Fgf21* (data not shown). However, a weak effect of treatment on methylation of cg21591807 was found, but this effect did not remain significant after adjustment for multiple testing (data not shown).

5) One of the novel aspects of these studies is the similar weight gain and fat mass change in the Drnmt3a adipose specific KO, compared to the control. Thus insulin sensitivity is maintained even in the face of greater adiposity, a key finding that needs more emphasis in the Results and Discussion.

We appreciate reviewer’s suggestion, and we have now emphasized the importance and uniqueness of this finding in the Discussion.

6) The authors fail to cite the paper by Kamei et al. in Obesity 2010, which demonstrated that an aP2 transgenic Dnmt3 mouse (PMID: 19680236) exhibit similar weight gain to wt mice; but importantly those mice have increased expression of Tnfa and MCP-1 in fat from their adipose depots. Similarly in that paper Dnmt3 expression was higher in WT mice fed a HFD. There should be a citation and also discussion about that paper (limitations based on the aP2 promoter, etc.).

We thank the reviewer for bringing up this reference, which we had overlooked. We now cite it. As noted, the authors of this paper generated adipose-selective Dnmt3a-overexpressor mice, which did not show dramatic changes in whole body metabolism on chow, HFD and nor with high methyl donor diet on chow. Of note, those authors did minimal physiological characterization of these mice, restricted mainly to a single time point of body weight and serum lipids, glucose and insulin. An additional caveat, also noted by the reviewer, is the use of the aP2 (Fabp4) promoter to drive transgene expression. As described in recent papers (Jeffery et al., 2014; Kang, Kong and Rosen, 2014; Lee et al., 2013), there are significant concerns about specificity as the aP2 gene is also expressed in non-adipose tissues including macrophages. Another potential compounding factor is the ‘*timing*’ of aP2 expression, which is speculated to be early during adipose development. Some studies have shown that Dnmt3a affects adipogenesis (Guo, Chen, Yang, Zhu and Wu, 2016), and so, a potential developmental effect of transgene expression might have confounded interpretation of the metabolic effect in this model. Per the reviewer’s suggestion, we investigated inflammation in Dnmt3a KO mice. We observe a non-significant trend toward reduced TNF and MCP-1 expression in KO mice (Figure 4—figure supplement 5). We also saw no obvious change in macrophage infiltration by histological examination in Dnmt3a KO mice after HFD (Figure 4—figure supplement 5).

Reviewer #2:[…] 1) What is the rationale for using the epididymal adipose tissue depot to measure HFD-mediated induction of Dnmt3a? Are there differences in HFD regulation of Dnmt3a between visceral and subcutaneous adipose tissue? Fgf21?

Epididymal adipose tissue is a large and accessible depot, and its mass correlates well with insulin resistance. We have now measured Dnmt3a expression in BAT, subcutaneous WAT, and mesenteric WAT from chow vs HFD mice (Figure 1—figure supplement A). Dnmt3a expression in all depots were significantly elevated but most prominently in visceral WAT (i.e. epididymal and mesenteric WAT).

2) Although not as dramatic as in adipose tissue, there is a relatively robust and significant increase of Dnmt3a in skeletal muscle in mice fed a HFD (Figure 1—figure supplement 1). What is the relative expression of skeletal muscle Dnmt3a with respect to adipose tissue and liver? What are the potential implications for HFD-mediated up-regulation of skeletal muscle Dnmt3a with respect to Fgf21 expression?

Based on the reviewer’s suggestion, we measured the relative expression of Dnmt3a between those three tissues. Protein expression of Dnmt3a is highest in epididymal fat compared to liver and muscle (Figure 4—figure supplement 1). It seems possible that Dnmt3a could suppress Fgf21 levels in skeletal muscle as it does in adipose tissue. Overall, however, serum FGF21 levels rise in response to a HFD (Figure 4—figure supplement 3).

Reviewer #3:[…] If the authors are able to address the point below, the manuscript would be further strengthened:1) In their studies here, the authors only look at Dnmt3a in gonadal white fat (eWAT). Since Dnmt3a is apparently broadly expressed, it would be interesting to know whether it is also increased in subcutaneous/beige fat and in brown fat in models of insulin resistance. Moreover, it would be relevant to know whether beige/brown fat are altered in the adipocyte-specific Dnmt3a knockout model. If for example, the mutant mouse has activated brown or beige fat, this could be an explanation for the improved glucose and insulin tolerance.

This is a fair point. We now add data showing how Dnmt3a expression changes in additional depots after HFD (Figure 1—figure supplement 1); the upshot is that there is a greater effect in visceral depots (epiWAT and mesenteric WAT) than in iWAT or BAT.

In addition, Dnmt3a KO mice do not display changes in body weight or adiposity at baseline or in response to fasting (Figure 4), suggesting that increased thermogenesis/energy expenditure is not a likely mechanism for improved insulin sensitivity in the KO mice. However, to rule out a subtle but significant change in beige/BAT activity, we assessed the expression of major thermogenic genes in subcutaneous WAT and BAT. We confirm there is no major impact on thermogenic genes in beige and brown fat in KO animals; if anything, we note a slight *reduction* of *Ucp1* expression in subcutaneous WAT (Figure 4—figure supplement 4). We also did not observe any obvious histological changes that would suggest increased browning (Figure 4—figure supplement 5).